# Clinical features and natural history of the first 2073 suspected COVID-19 cases in the Corona São Caetano primary care programme: a prospective cohort study

Fabio E Leal,[1,2] Maria C Mendes-Correa,[3] Lewis Fletcher Buss [iD],[3] Silvia F Costa,[3] Joao C S Bizario,[1] Sonia R P de Souza,[1] Osorio Thomaz,[4] Tania Regina Tozetto-Mendoza,[3] Lucy S Villas-Boas,[3] Léa Campos de Oliveira-da-Silva,[5] Regina M Z Grespan,[1] Ligia Capuani,[6] Renata Buccheri,[3] Helves Domingues,[7] Neal Alexander,[8] Philippe Mayaud,[8] Ester Cerdeira Sabino[3]

► Prepublication history and supplemental material for this paper is available online. To view these files, please visit the journal online (http://dx.doi.org/10.1136/bmjopen-2020-042745).

FEL, MCM-C and LFB contributed equally.

For numbered affiliations see end of article.

**Correspondence to**
Dr Lewis Fletcher Buss;
lewisbuss@usp.br

## ABSTRACT

**Background** Despite most cases not requiring hospital care, there are limited community-based clinical data on COVID-19.

**Methods** The Corona São Caetano programme is a primary care initiative providing care to all residents with COVID-19 in São Caetano do Sul, Brazil. It was designed to capture standardised clinical data on community COVID-19 cases. After triage of potentially severe cases, consecutive patients presenting to a multimedia screening platform between 13 April and 13 May 2020 were tested at home with SARS-CoV-2 reverse transcriptase (RT) PCR; positive patients were followed up for 14 days with phone calls every 2 days. RT-PCR-negative patients were offered additional SARS-CoV-2 serology testing to establish their infection status. We describe the clinical, virological and natural history features of this prospective population-based cohort.

**Findings** Of 2073 suspected COVID-19 cases, 1583 (76.4%) were tested by RT-PCR, of whom 444 (28.0%, 95% CI 25.9 to 30.3) were positive; 604/1136 (53%) RT-PCR-negative patients underwent serology, of whom 52 (8.6%) tested SARS-CoV-2 seropositive. The most common symptoms of confirmed COVID-19 were cough, fatigue, myalgia and headache; whereas self-reported fever (OR 3.0, 95% CI 2.4 to 3.9), anosmia (OR 3.3, 95% CI 2.6 to 4.4) and ageusia (OR 2.9, 95% CI 2.3 to 3.8) were most strongly associated with a positive COVID-19 diagnosis by RT-PCR or serology. RT-PCR cycle thresholds were lower in men, older patients, those with fever and arthralgia and closer to symptom onset. The rates of hospitalisation and death among 444 RT-PCR-positive cases were 6.7% and 0.7%, respectively, with older age and obesity more frequent in the hospitalised group.

**Conclusion** COVID-19 presents in a similar way to other mild community-acquired respiratory diseases, but the presence of fever, anosmia and ageusia can assist the specific diagnosis. Most patients recovered without requiring hospitalisation with a low fatality rate compared with other hospital-based studies.

### Strengths and limitations of this study

► The clinical features of COVID-19 have mostly been described in hospital-based studies which are biased towards severe disease.
► We report a prospective cohort of suspected and confirmed COVID-19 cases from a primary care initiative in the Brazilian municipality of São Caetano do Sul.
► By systematically testing consecutive suspected community cases with molecular and serological tests, we were able to address the diagnostic value of clinical features of mild-to-moderate COVID-19 in primary care.
► Prospective follow-up of confirmed cases and linkage with hospital datasets allowed us to describe the natural history of a primary care COVID-19 population.
► A limitation of the work was that not all PCR-negative participants underwent serology testing due to loss to follow-up.

## INTRODUCTION

A comprehensive public health response is vital but difficult to achieve during an epidemic. The COVID-19 pandemic, caused by the novel SARS-CoV-2, started in China in late 2019.[1] According to the World Health Organization (WHO)[2 3] and others,[4 5] the ideal early response should have been multipronged, with identification, isolation, treatment and contact tracing of symptomatic cases, relying on a strong testing programme. Primary healthcare (PHC) is well placed to implement such a response, by identifying cases early and managing them in a way that minimises overcrowding of emergency rooms and intensive care units.[6 7] Real-time data

analysis coming from these primary care response systems can inform policy decisions.

PHC in Brazil is provided by the publicly funded Unified Health System (SUS—Portuguese acronym) within the family health strategy (*Estratégia Saúde da Família*). Provision of care is centred around a healthcare unit with a multiprofessional team that is responsible for all residents in the immediate catchment area.[8] Nearly two-thirds of the Brazilian population is covered by the family health strategy.[8]

In Brazil, the first case of COVID-19 was identified in the city of São Paulo on 26 February 2020.[9] As of 1 December 2020, there were over 6 million confirmed cases nationally, with São Paulo contributing a fifth of these.[10] The reasons for the exceptionally large epidemic in Brazil have been discussed elsewhere.[11–13] In March 2020, the Municipal Health Department of the municipality of São Caetano do Sul—part of the Greater Metropolitan Region of São Paulo—began to develop a clinical and testing platform to organise its COVID-19 response. The aim was to provide universal detection and management of symptomatic cases and their contacts. The platform was developed in partnership with two local universities—the Municipal University of São Caetano do Sul (USCS) and the University of Sao Paulo—and called 'Corona São Caetano'.

Large-scale community-based observational cohorts are difficult to establish under epidemic circumstances, particularly if the risk of exposure for research personnel is high. Hence, most COVID-19 epidemiological and clinical studies have been hospital-based,[14–16] and therefore tend to include more severe cases whose findings may not be generalisable to the general population,[17] although some limited descriptions from ambulatory settings are available.[18–20] The objectives of this study were to describe the epidemiological indicators of the early phase of the programme rollout; and to describe the clinical, virological and natural history features (including hospitalisation and deaths) of SARS-CoV-2 infection among patients identified in primary care.

## METHODS
### Setting
The municipality of São Caetano do Sul has a population of 161 000 inhabitants.[21] The city's population is older than the Brazilian population[21] and its Human Development Index is one of the highest in the country. Nearly all (97.4%) children aged 6–14 years are in education and 31% of the population have completed higher education[22] (Brazilian national average is 11%).

### Corona São Caetano platform
The objective of the platform was to offer clinical care for patients with influenza syndrome and suspected COVID-19. Through the multimedia platform (website or phone call), patients could be triaged and guided in relation to their clinical needs and tested, without having to leave

their homes or go to health facilities, unless seriously ill. This strategy aimed at reducing the workload in health units and the risk of SARS-CoV-2 transmission in the population served by these health units. Patients' general practitioners (GPs) were informed of lab results and had access to clinical data stored in the platform. GPs were expected to call patients being assisted by the platform and provide medical assistance through home visits or at the primary care clinic if needed. In general, the drugs prescribed through the platform were restricted to analgesics and antipyretics. The platform was designed so that clinical information was collected in a standardised way for research purposes.

Residents of the municipality aged 12 years and older with suspected COVID-19 symptoms were encouraged, through local media reports, to contact the dedicated Corona São Caetano platform via the website or by phone. They were invited to complete an initial screening questionnaire that included sociodemographic data; information on symptoms type, onset and duration; and recent contacts.

Patients meeting the suspected COVID-19 case definition (ie, having at least two of the following symptoms: fever, cough, sore throat, coryza or change in/loss of smell (anosmia); or one of these symptoms plus at least two other symptoms consistent with COVID-19) were further evaluated, while people not meeting these criteria were reassured, advised to stay at home and contact the service again if they were to develop new symptoms or worsening of current ones. The case definition was developed in consultation with infectious disease and primary care specialists to encompass the known symptoms of COVID-19 and is similar to the Brazilian national case definition.[23] Patients were then called by a medical student to complete a risk assessment. All pregnant women, and patients meeting predefined triage criteria for severe disease (see online supplemental material 1), were advised to attend a hospital service—either an emergency department or outpatient service, depending on availability. All other patients were offered a home visit for self-collection of a nasopharyngeal swab (NPS).

### Sample collection
Patients self-collected NPS (both nostrils and throat) at their own homes under the supervision of trained healthcare personnel. We sent a link to an instructional video (https://youtu.be/rWZzV2ZP7KY) before the home visit to provide guidance on self-collection procedures. NPS for the molecular detection of SARS-CoV-2 has been recommended as an alternative method of collection for samples from patients with suspected COVID-19,[24] as well as other respiratory diseases, and has the advantage of reducing the chance of aerosol transmission to healthcare professionals. Healthcare personnel were instructed to maintain a distance of 6 ft from the patient and to wear personal protective equipment at all times. Samples were immediately put on a cool box between 2°C–8°C and

stored at 4°C in a fridge until shipment to the lab within 24 hours.

## Follow-up procedures

Patients testing SARS-CoV-2 RT-PCR positive were followed up to 14 days[25] (a maximum of seven phone calls) from completion of their initial questionnaire. They were contacted every 48 hours by a medical student who completed another risk assessment and recorded any ongoing or new symptoms. The purpose of the follow-up was to assess clinical evolution. Where patients were judged to be deteriorating or developing severe disease they were signposted to secondary care services. Patients testing RT-PCR negative were followed up by the PHC programme for their residential area. They were advised to contact the platform for a new consultation if they developed new symptoms. Starting on 19 May, when serological testing became available, RT-PCR-negative patients were recontacted to offer antibody (IgG/IgM combined) testing 14 days after their initial registration as long as they had become asymptomatic.

## Study dates

The Corona São Caetano programme was launched on 6 April 2020, with a 1 week pilot phase designed to test instruments before roll-out. For this analysis, we included all patients making their first contact with the programme in its first month, that is, between 13 April and 13 May 2020. The period of follow-up (last date of data extraction) was 4 June 2020, to account for the accrual period (3 weeks) of possible hospitalisations in the last included patients.

## Laboratory methods

Due to shortages of some reagents, we used two RT-PCR platforms at different times during the study: ALTONA RealStar SARS-CoV-2 RT-PCR Kit 1.0 (Hamburg, Germany) and the Mico BioMed RT-qPCR kit (Seongnam, South Korea). For serology we tested 10 μL of serum or plasma (equivalent in performance) using a qualitative rapid chromatographic immunoassay (Wondfo Biotech, Guangzhou, China), which jointly detects anti-SARS-CoV-2 IgG/IgM. The assay has been found to have a sensitivity of 81.5% and specificity of 99.1% in a US study.[26] In our local validation, after 2 weeks of symptoms, the sensitivity in 59 RT-PCR confirmed cases was 94.9%, and specificity in 106 biobank samples from 2019 was 100%.

## Statistical methods

We estimated the contribution of our platform to total number of COVID-19 cases diagnosed in São Caetano do Sul. To do this, we compared the number of cases diagnosed in our programme with official data released by the Municipal Department of Health in its daily bulletins (https://coronavirus.saocaetanodosul.sp.gov.br).

Clinical and demographic data were extracted directly from the Corona São Caetano information system. To analyse clinical presentation, we first calculated the proportion and exact binomial 95% CIs of cases reporting each symptom in the three testing groups: SARS-CoV-2

RT-PCR positive; RT-PCR negative/seropositive and RT-PCR negative/seronegative. We next combined RT-PCR and serology positive cases to make a confirmed COVID-19 group, and those negative on both tests to make a SARS-CoV-2-negative control group. We express the association between each symptom and a positive COVID-19 diagnosis as ORs and 95% CIs.

Next, we assessed associations between RT-PCR cycle thresholds (Cts) and other clinical features. ALTONA and MiCo BioMed RT-PCR kits each separately amplify two different SARS-CoV-2 viral genes, as such each patient had two Ct values. There was a high concordance between Cts for the two genes within each kit (online supplemental figure S1), and we opted therefore to use the mean of the two Ct values for each patient in all analyses. We calculated univariable associations between Cts and age, sex, delay from symptom onset to NPS collection and presenting symptoms using simple linear regression. We then built a multivariable linear regression model to assess independent associations between presenting symptoms and RT-PCR Cts. As age, sex and time of swab collection may confound this relationship, we included these variables, as well as the RT-PCR platform (ALTONA vs MiCo BioMed), as covariates in the model.

For RT-PCR-positive patients, hospitalisations and deaths were extracted from the study platform. To extend the follow-up period and to capture RT-PCR-negative patients and those initially triaged to hospital (no study follow-up), hospitalisation and vital status was confirmed by linkage with two administrative databases: the municipal epidemiological surveillance dataset, as well as the state-wide influenza-like illness notification system (SIVEP-Gripe). Linkage was last performed on 5 June 2020, 23 days after the last patient was enrolled, by the author SRPS who did not have access to the full analytic dataset. This author searched the SIVEP-Gripe system and the municipal epidemiological surveillance dataset using full name and date of birth. Categorical patient characteristics were compared between patients requiring and those not requiring hospitalisation using a $\chi^2$ test or Fisher's exact test. Continuous variables were compared using the Wilcoxon rank sum test. A multivariate analysis was not conducted due to the small number of individuals experiencing this outcome.

The cohort sample included consecutive cases presenting to the Corona São Caetano programme and a formal sample size calculation was not performed. Missing data were excluded. All analyses were conducted in R Software for Statistical Computing, V.3.6.3.[27]

## Patient and public involvement

Patients were not involved in the planning of this research.

## RESULTS
### Epidemiological and programmatic indicators

Over the study period, there were 2073 presentations (49% phone call, 51% website), from 2011 individual

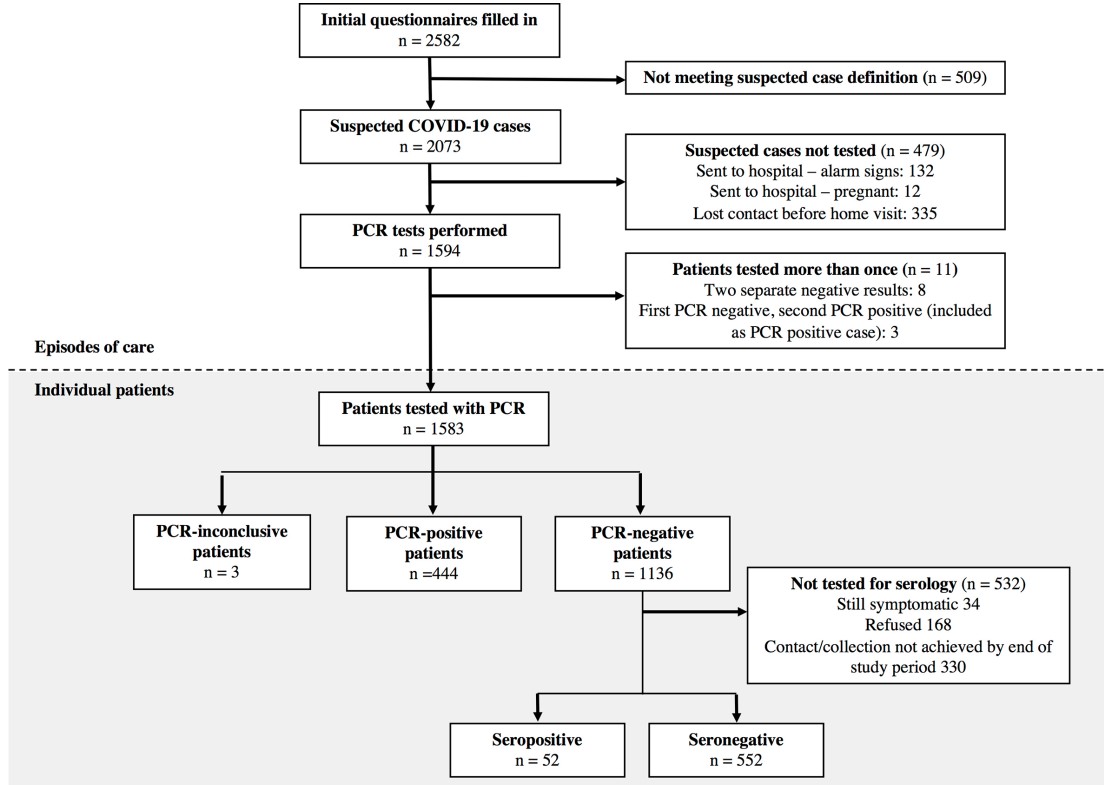

**Figure 1** Patient flow chart for the Corona São Caetano platform between 13 April and 13 May 2020. In the upper section (white background), the numbers correspond to individual presentations to the system; among 2073 suspected cases, 60 had two presentations and 1 had three. In the lower section (grey background), numbers correspond to individual patients making up the final analytic groups.

patients, that met the criteria for a suspected COVID-19 case (see figure 1 for study flow). At initial phone interview, 132 (6%) potential cases were advised to go directly to a health service based on the triage questions and 12 (0.6%) because of pregnancy. Only four (3%) of referred patients were admitted to hospital and none died.

In total, 1583 individual patients were tested with RT-PCR for SARS-CoV-2; 444 (28.0%, 95% CI 25.9 to 30.3) were positive. The proportion of positive results was stable over the study (online supplemental figure S2). Among the RT-PCR-negative group, 604 (53% of 1136) underwent serology testing, of whom 52 (8.6%, 95% CI 6.6 to 11.1) were seropositive. The median (IQR) time from symptom onset to serology collection was 31 (26–37) days. The age-sex structure of patients being tested differed from the underlying population of São Caetano do Sul (online supplemental figure S3) with an over-representation of working-age adults and women. At the beginning of programme role out, 25% of notified COVID-19 cases in São Caetano do Sul were diagnosed in our programme. Over the study period, adherence to the programme increased, and by 13 May 2020, this figure had risen to 78%.

Of 444 RT-PCR-positive patients eligible for longitudinal follow-up, 326 (73%) had their final follow-up visit at least 14 days after their initial presentation. Of the seven possible follow-up questionnaires, 384 (86%)

patients with COVID-19 completed three or more, and 162 (36%) completed all seven.

## Participant characteristics

Patient characteristics are shown in table 1. Although women were over-represented in the cohort, there were proportionally more males in the RT-PCR-positive and seropositive groups compared with the seronegative group. Of note, 55% of RT-PCR-negative/seronegative patients had completed higher education compared with 35% RT-PCR-positive patients (p<0.001, $\chi^2$ test). The median number of days from symptom onset to swab collection was 5.0 (IQR 4.0–7.0) among RT-PCR-positive patients and 6.0 (IQR 4.0–8.3) among RT-PCR-negative/seropositive patients (p=0.06, Wilcoxon rank sum) (online supplemental figure S4). Chronic respiratory disease was less frequent in RT-PCR-positive than dual-negative patients.

## Symptoms of COVID-19

The prevalence of individual symptoms at presentation is shown in figure 2A stratified by final diagnostic category. The most frequent symptoms among RT-PCR and seropositive patients were headache (82% and 75%), myalgia (80% and 80%), cough (77% and 63%) and fatigue (77% and 79%). Anosmia was present in 56% and 63% of RT-PCR-positive and seropositive patients, respectively,

**Table 1** Demographic and clinical characteristics of 1048 suspected COVID-19 cases undergoing diagnostic testing in the Corona São Caetano programme

| | RT-PCR +ve (G1) n=444 n (%) or median (IQR) | RT-PCR -ve Sero +ve (G2) n=52 n (%) or median (IQR) | RT-PCR -ve Sero -ve (G3) n=552 n (%) or median (IQR) | P value G1 vs G2 | P value G1 vs G3 |
|---|---|---|---|---|---|
| **Sex** | | | | | |
| Male | 200 (45.0) | 23 (44.2) | 185 (33.5) | | |
| Female | 244 (55.0) | 29 (55.8) | 367 (66.5) | 1.0 | <0.001 |
| **Age groups (years)** | | | | | |
| 10–19 | 29 (6.5) | 1 (1.9) | 25 (4.5) | | |
| 20–39 | 197 (44.4) | 17 (32.7) | 236 (42.8) | | |
| 40–59 | 158 (35.6) | 28 (53.8) | 218 (39.5) | | |
| 60+ | 60 (13.5) | 6 (11.5) | 73 (13.2) | 0.07 | 0.40 |
| **Educational level** | | | | | |
| Up to primary education | 75 (16.9) | 7 (13.5) | 56 (10.2) | | |
| High school | 214 (48.3) | 19 (36.5) | 194 (35.2) | | |
| University | 154 (34.8) | 26 (50.0) | 301 (54.6) | 0.10 | <0.001 |
| **Essential occupation** | | | | | |
| Non-HCW essential job* | 137 (30.9) | 12 (23.1) | 148 (26.9) | | |
| Carers | 10 (2.3) | 0 (0.0) | 8 (1.5) | | |
| HCW | 32 (7.2) | 5 (9.6) | 73 (13.2) | | |
| No | 264 (59.6) | 35 (67.3) | 322 (58.4) | 0.45 | 0.01 |
| **Body mass index (kg/m$^2$)** | | | | | |
| <25 | 151 (34.2) | 22 (42.3) | 211 (38.4) | | |
| 25–29 | 182 (41.2) | 17 (32.7) | 187 (34.0) | | |
| 30–35 | 79 (17.9) | 9 (17.3) | 112 (20.4) | | |
| 35+ | 30 (6.8) | 4 (7.7) | 40 (7.3) | 0.62 | 0.14 |
| **Comorbidities** | | | | | |
| Cardiovascular disease | 88 (20.4) | 9 (17.6) | 129 (24.0) | 0.89 | 0.40 |
| Diabetes mellitus | 48 (11.1) | 4 (7.8) | 39 (7.3) | 0.86 | 0.12 |
| Any chronic respiratory disease | 37 (8.9) | 9 (18.0) | 79 (15.3) | 0.13 | 0.01 |
| COPD | 24 (5.5) | 5 (9.8) | 54 (10.1) | 0.47 | 0.03 |
| Chronic kidney disease | 1 (<1) | 0 (0.0) | 3 (1.0) | 1.0 | 0.83 |
| Time from symptom onset to swab collection (days), median (IQR) | 5.0 (4.0–7.0) | 6.0 (4.0–8.3) | 6.0 (4.0–9.0) | 0.06 | <0.001 |

Missing data—educational level 2; essential occupation 2; body mass index 4; cardiovascular disease 28; diabetes 31 mellitus; chronic respiratory disease 65; chronic kidney disease 27; COPD 28.

P values calculated by $\chi^2$ test, Fisher's exact test or Wilcoxon rank sum test.

*Security, emergency services, supermarket, public transport and pharmacy workers.

COPD, chronic obstructive pulmonary disease; HCW, healthcare workers; RT, reverse transcriptase.

compared with 30% in those testing doubly negative. A similar pattern was observed for ageusia (53% and 53% vs 30%). Upper respiratory tract symptoms—including coryza, blocked nose, ageusia and anosmia—were more frequent in younger people (figure 2B). Symptoms were similar in men and women (figure 2C). The evolution of symptoms over time among RT-PCR-positive patients is shown in online supplemental figure S5.

The ORs for testing positive for SARS-CoV-2 (RT-PCR or serology) associated with each presenting symptom are shown in figure 3. The symptoms with strongest associations were anosmia (OR 3.3, 95% CI 2.6 to 4.4), fever (3.0, 95% CI 2.4 to 3.9) and ageusia (2.9, 95% CI 2.3 to 3.8). The presence of sore throat (0.53, 95% CI 0.41 to 0.68) and diarrhoea (0.72, 95% CI 0.55 to 0.96) were associated with a negative SARS-CoV-2 test.

### Associations between SARS-CoV-2 RT-PCR cycle threshold (Ct) values, and demographic and clinical features

Figure 4 shows the associations between mean RT-PCR cycle threshold and demographic features and symptoms at presentation (the median (IQR) time from

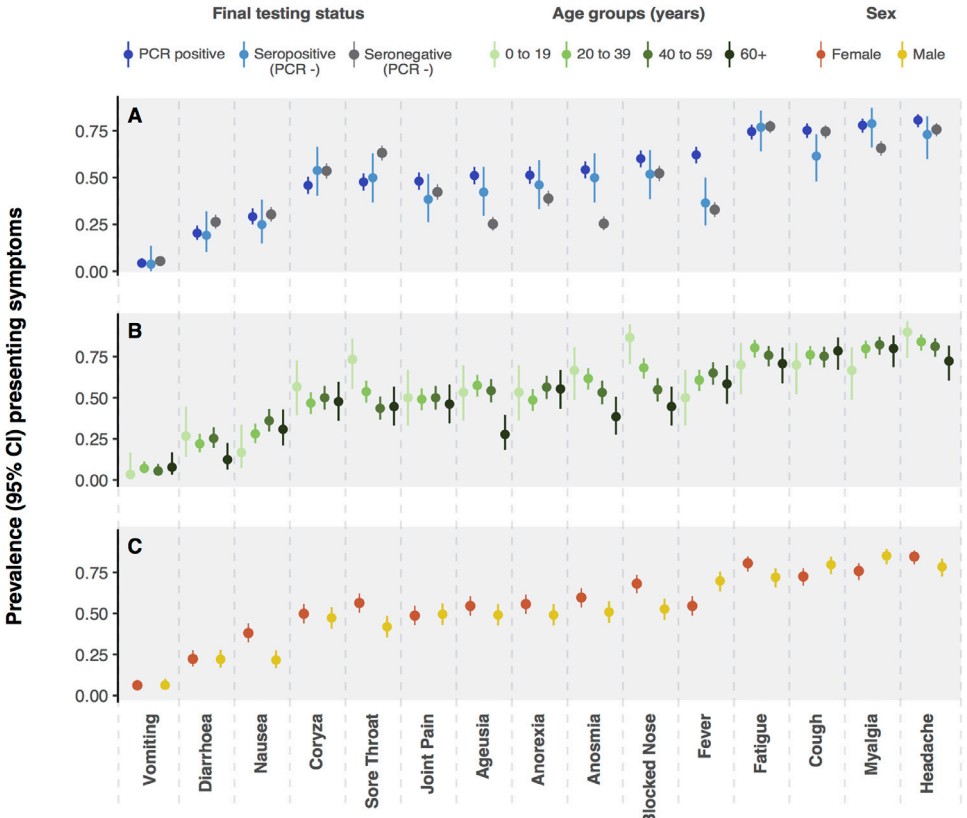

**Figure 2** Panel A presents prevalence (point) and exact binomial 95% CIs (vertical lines) of symptoms at presentation among patients with suspected COVID-19 according to reverse transcriptase (RT)-PCR result and serostatus (A). Panels B and C present the prevalence of presenting symptoms among patients with COVID-19 (RT-PCR and serology positive) stratified by age (B) and sex (C).

presentation to swab was 1 (1–2) day). Older age was associated with lower cycle thresholds, with a change in mean Ct of –0.05 (95% CI –0.09 to –0.01) for each additional year of age (figure 4B). The mean difference in Ct value was –1.36 (95% CI –2.49 to –0.23) in men compared with women (figure 4C). For each doubling in the number of days from symptom onset to swab collection the mean Ct

value increased by 3.28 (95% CI 2.33 to 4.03) (figure 4A). Presenting symptoms of fever and arthralgia were associated with lower Cts, whereas anosmia, ageusia, vomiting, diarrhoea and nausea were associated with higher Cts (figure 4D and online supplemental table S1). After adjustment for age, sex, delay from symptom onset and RT-PCR platform used, fever (–0.06, 95% CI –2.11 to –0.001) and arthralgia (–1.24, 95% CI –2.18 to –0.10) remained associated with lower Cts, and anosmia (2.21, 95% CI 1.0 to 3.29), ageusia (1.96, 95% CI 0.88 to 3.0) and diarrhoea (1.36, 95% CI 0.12 to 2.61) with higher Cts (online supplemental table S1).

### Hospitalisations and deaths
Of the 444 RT-PCR-positive patients, 30 (6.8%) had been hospitalised by 5 June 2020, when the database linkage was last updated, and 3 (0.7%) had died; in-hospital mortality was therefore 10% (3/30). In 28 cases the date of admission was available. The median time from symptom onset to hospital admission was 7 (range 2–14) days. Among 1136 RT-PCR-negative patients, 6 (0.5%) had been admitted to hospital. One (<0.01% of 1,136) of these six patients died. None of the 604 RT-PCR-negative patients who underwent serology was admitted to hospital or died. Table 2 compares patient characteristics by hospitalisation status. Notably, hospitalised patients were older,

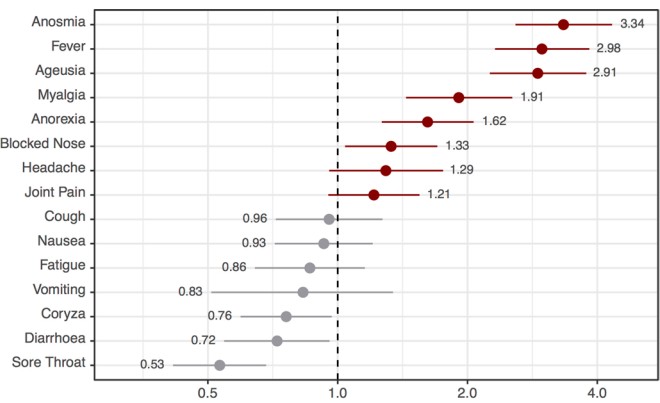

**Figure 3** ORs (black dot) and 95% CIs (lines) for testing positive for COVID-19 (reverse transcriptase (RT)-PCR positive or serology positive) associated with the presence of each presenting symptom. Horizontal axis is on log scale. Point estimates of ORs are shown inline with their corresponding symptom.

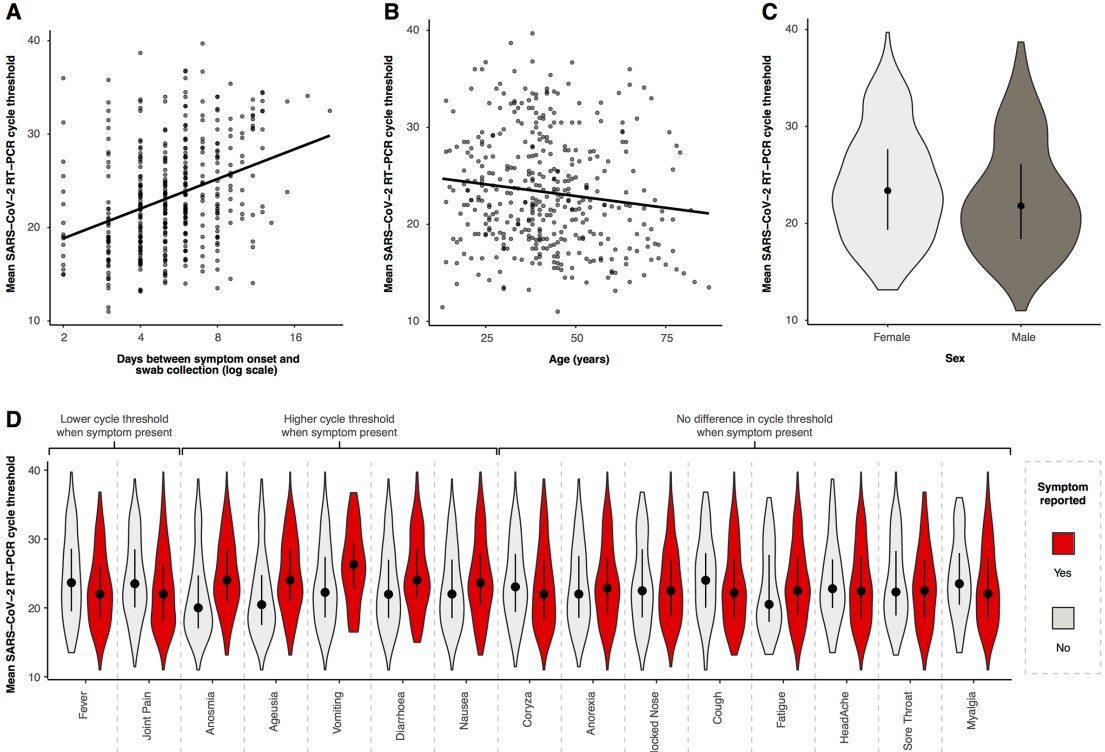

**Figure 4** Relationship between mean reverse transcriptase (RT)-PCR cycle threshold (Ct) and day of illness course when the nasopharyngeal swab was collected (A), patient age (B), patient sex (C) and different symptoms at presentation (D). Panels A and B show the best-fit linear regression lines, panels C and D are violin plots (rotated kernel density plots showing the full distribution of data) of the Ct values with median (black dot) and IQR (black line).

had more cardiovascular comorbidities and were more frequently obese.

## DISCUSSION

We present a community-based cohort of suspected COVID-19 cases recruited through a primary care initiative in the Brazilian municipality of São Caetano do Sul. Offering RT-PCR testing to all patients presenting with symptoms compatible with COVID-19, the positivity rate was 28%, with 8.6% of those testing negative subsequently found to be seropositive, that is, >35% of the cohort had a diagnosis of COVID-19. Anosmia, ageusia and self-reported fever provided the greatest diagnostic value in identifying COVID-19. The rate of hospitalisation and deaths among RT-PCR-positive patients was low, at 6.8% and 0.7%, respectively. Our results provide important information on the clinical presentation, diagnostic testing and natural history of COVID-19 identified in the community.

The profile of suspected cases that tested positive for COVID-19 differed in some important respects from those testing negative. The lower educational level among positive cases suggests that, in São Caetano do Sul, the risk of exposure to COVID-19 follows a socioeconomic gradient, consistent with other findings from Brazil.[13 28] Although more women presented to the platform, proportionally more men tested positive, consistent with data from São Paulo showing a higher seroprevalence in men than women,[11] and potentially reflecting

different health-seeking behaviours. Comorbidities were mostly similar, although chronic respiratory disease was less frequent in those testing RT-PCR positive. This may be due to a proportion of presentations in those with chronic respiratory disease being explained by exacerbations of their underlying pathology from aetiologies other than SARS-CoV-2, as well as higher anxiety about COVID-19 in those with pre-existing respiratory disease.

Extrapolating the seropositivity rate among RT-PCR-negative patients to the 532 who were not tested with serology, we estimate that an additional 46 seropositive cases would have been identified. As such, 18% (98/542) of COVID-19 cases were missed by RT-PCR in the setting of symptomatic presentations to primary care. This is similar to a pooled analysis showing a false-negative rate for RT-PCR of 20% at 3 days postsymptom onset.[29] Viral load peaks around the time of symptom onset and remains high over the first symptomatic week (figure 4A).[30 31] Consistent with this, we found a slightly longer delay to swab collection (due to delay in presentation to the platform) in RT-PCR false-negative patients than RT-PCR positive patients (online supplemental figure S4).

COVID-19 presents in a similar way to other respiratory viral illnesses. Indeed, in our cohort the most common symptoms of COVID-19—such as cough, fatigue, headache and so on—were reported with a similar frequency among patients testing negative. It is therefore important to have identified anosmia, ageusia, self-reported fever, myalgia and anorexia as the symptoms with greatest value in the differential diagnosis

**Table 2** Characteristics of RT-PCR-positive patients stratified by hospitalisation status

| | Hospitalised n=30 n (%) or median (IQR) | Not hospitalised n=414 n (%) or median (IQR) | P value |
|---|---|---|---|
| **Age (years)** | | | |
| 10–19 | 1 (3) | 28 (97) | |
| 20–39 | 6 (3) | 191 (97) | |
| 40–59 | 14 (9) | 144 (91) | |
| 60+ | 9 (15) | 51 (85) | 0.006 |
| **Sex** | | | |
| Female | 16 (7) | 228 (93) | |
| Male | 14 (7) | 186 (93) | 0.852 |
| **Comorbidities** | | | |
| Cardiovascular disease | 11 (13) | 77 (87) | 0.001 |
| Diabetes mellitus | 8 (17) | 40 (83) | 0.007 |
| Any chronic respiratory disease | 2 (5) | 35 (95) | 1.0 |
| COPD | 1 (5) | 23 (95) | 1.0 |
| Chronic kidney disease | 1 (100) | 0 (0) | 0.06 |
| **Body mass index (kg/m$^2$)** | | | |
| <25 | 4 (3) | 147 (97) | |
| 25–29 | 8 (4) | 174 (96) | |
| 30–35 | 12 (15) | 67 (85) | |
| 35+ | 6 (20) | 24 (80) | <0.001 |
| Time to presentation (days) | 3 (3–4) | 4 (3–5) | 0.072 |

Missing data—body mass index 2; cardiovascular disease 12; diabetes mellitus 12; chronic respiratory disease 29; COPD 11; chronic kidney disease 12.

COPD, chronic obstructive pulmonary disease; RT, reverse transcriptase.

of COVID-19 in primary care. This is consistent with systematic review evidence highlighting anosmia and ageusia as key diagnostic features of COVID-19.[32] It is of note that 30% of jointly RT-PCR and serology negative patients reported these symptoms, indicating that although indicative of COVID-19, the specificity of these symptoms is not high enough to rule in the diagnosis alone. Sore throat and diarrhoea—both considered symptoms of COVID-19 in other settings[33]—were more frequently due to other possible aetiologies in this primary care context.

These results are robust for a number of reasons. First, our sample is representative of the population of interest—that is, consecutive patients with suspected COVID-19 in the community—instead of extrapolating from hospital cases. Symptom data were collected prospectively, eliminating recall or interviewer bias. Finally, we have a control group of patients who were negative for both RT-PCR and serology, minimising misclassification due to false-negative RT-PCR.

In our study, the proportion of patients with a positive SARS-CoV-2 RT-PCR requiring hospitalisation was low (7%). Early reports from China were of 13.8% of cases being severe,[34] but this value was lower when under-ascertainment of cases was accounted for.[35 36] This is because our cohort reflects mild-to-moderate cases, as severely ill patients are likely to have attended hospital

directly. As such, only 3% of patients we triaged to attend health services were ultimately hospitalised, possibly due to self-selection of patients presenting to our service. Supporting this, our overall case fatality ratio among RT-PCR-positive patients was 0.7%. The rate of hospitalisation was lower (0.5%) in those testing PCR-negative. These patients were admitted with a severe acute respiratory syndrome of an aetiology other than SARS-CoV-2. The 14-fold higher admission rate among PCR-positive cases highlights the importance of molecular testing for SARS-CoV-2 in patients presenting with features of respiratory viral illness to primary care.

As expected, the main determinant of Ct was the delay between symptom onset and swab collection, mostly due to the delay in reporting to the platform. After adjusting for this, as well as age and sex, we found that a self-reported fever and arthralgia were associated with lower Cts. The presence of these symptoms may identify patients with a higher viral load in the community. However, these results should be seen as purely exploratory, and the widespread Ct values around the regression line precludes a direct clinical application at present.

Our study has some limitations. First, the small sample size preluded a multivariate analysis of factors associated with hospitalisation or death. Next, serology was not performed on

all RT-PCR-negative patients due to ongoing symptoms, loss to follow-up or patient refusal. Of note, none of the RT-PCR-negative patients who were admitted to hospital underwent serology testing. This suggests that patients who were not tested with serology may have had a higher prevalence of COVID-19 than those that were tested. In addition, imperfect serology test performance (81% sensitivity)[26] will introduced false-negative results. Taken together, these biases may have underestimated the true seroprevalence among RT-PCR-negative cases, as well as the false-negative rate of RT-PCR. The latter calculation may also have been influenced by the inclusion of RT-PCR-positive patients in the denominator, introducing an incorporation bias.[37] Furthermore, the association between symptoms and COVID-19 diagnosis was based on the comparison with doubly PCR and serology negative individuals. It is not clear how the exclusion of individuals that did not undergo serology testing would have influenced these associations. Finally, patients were not involved in the planning of the Corona platform or the research proposal.

A key strength to our study relates to the provision of PHC in Brazil and its symbiosis with medical training nationwide. PHC—within the family health strategy (*Estratégia Saúde da Família*)—is centred around a healthcare unit with a multiprofessional team that is responsible for all residents in the immediate catchment area.[8] São Caetano do Sul has enough GP units within the family health strategy that all residents have access to primary care. Medical students from the municipal university (USCS) are integrated into the PHC teams and progressively trained from the first year of medical school. Our initiative took advantage of this existing system, with the addition of an online platform allowing remote clinical assessment and follow-up. The suspension of normal clinical training at the medical school provided the workforce. The partnership with the University of São Paulo, which provided the laboratory diagnostics, created the unique opportunity to establish our prospective community cohort of suspected and confirmed COVID-19 cases. But we believe that this infrastructure could be implemented in other regions with less resources. Other respiratory disease such as influenza, measles or tuberculosis may benefit from similar approach. However, further evaluation of the impact of the Corona platform are required.

## CONCLUSION

Systematic testing of all suspected COVID-19 cases was feasible within primary care services in a Brazilian municipality. Anosmia, agueusia and fever provide the greatest diagnostic discrimination from other similar primary care presentations. Home care is a valid approach for most of these patients with a low rate of hospitalisation and death.

Our programme model—integrating multimedia technology, telehealth with universal access to primary care—may be successful in other contexts.

**Author affiliations**
¹Faculdade de Medicina, Universidade de São Caetano do Sul, São Paulo, Brazil
²Programa de Oncovirologia, Instituto Nacional de Câncer, Rio de Janeiro, Brazil
³Instituto de Medicina Tropical (LIM-52, LIM-46, LIM-49) and Departamento de Moléstias Infecciosas e Parasitárias, Universidade de São Paulo, São Paulo, Brazil
⁴Instituto de Pesquisas Tecnológicas, São Paulo, Brazil
⁵Laboratório de Medicina Laboratorial (LIM03), Hospital das Clínicas, Faculdade de Medicina, Universidade de São Paulo, São Paulo, Brazil
⁶Department of Services and Systems Design, Modular Research System Ltda, São Paulo, Brazil
⁷Department of Information Technology, Modular Research System Ltda, São Paulo, Brazil
⁸Faculty of Infectious & Tropical Diseases, London School of Hygiene and Tropical Medicine, London, UK

**Contributors** FEL, MCM-C, SFC, RB and ECS conceived and designed the study. FEL, RMZG and JCSB provided clinical oversight and supervision of medical students. FEL, MCM-C, LFB, HD, OT, LC and SRPS collected and curated the data. MCM-C, TRT-M, LSV-B and LCO-S performed the laboratory analysis. LFB performed the formal statistical analysis with assistance from FEL, SRPS, NA, PM, ECS and OT. LFB, FEL, PM and ECS wrote the first draft, and all authors reviewed, contributed to and approved the final version.

**Funding** The municipal health department of São Caetano do Sul (Secretaria Municipal de Saúde da Prefeitura de São Caetano do Sul) funded the establishment and implementation of the platform. The authors would like to acknowledge an award from FAPESP (2018/14389-0) and the UK Medical Research Council (MR/S0195/1) to the Brazil-UK Centre for Arbovirus Discovery, Diagnosis, Genomics and Epidemiology (CADDE).

**Competing interests** None declared.

**Patient consent for publication** Not required.

**Ethics approval** The study was approved by the local ethics committee (Comissão de Ética para Análise de Projeto de Pesquisa, protocol no. 13915, dated 3 June 2020). The committee waived the need for informed consent and allowed the development of an analytical dataset with no personal identification for the current analysis.

**Provenance and peer review** Not commissioned; externally peer reviewed.

**Data availability statement** Data are available in a public, open access repository: https://figshare.com/articles/dataset/Clinical_features_and_natural_history_of_the_first_2_073_suspected_COVID-19_cases_in_the_Corona_S_o_Caetano_primary_care_programme_a_prospective_cohort_study/13322474.

**ORCID iD**
Lewis Fletcher Buss http://orcid.org/0000-0002-9009-9301

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
