## [Reviewer comments · BMJ Open]

ARTICLE DETAILS

TITLE (PROVISIONAL)	Clinical features and natural history of the first 2,073 suspected COVID-19 cases in the Corona São Caetano primary care programme: a prospective cohort study
AUTHORS	Leal, Fabio; Mendes-Correa, Maria; Buss, Lewis; Costa, Silvia; Bizario, Joao; de Souza, Sonia; Thomaz, Osorio; Tozetto-Mendoza, Tania; Villas-Boas, Lucy; Oliveira, Lea; Grespan, Regina; Capuani, Ligia; Buccheri, Renata; Domingues, Helves; Alexander, Neal; Mayaud, Philippe; Sabino, Ester

VERSION 1 – REVIEW

REVIEWER	Stefan Morreel University of Antwerp Belgium
REVIEW RETURNED	18-Sep-2020

GENERAL COMMENTS	General remarks: This article with a simple design is relevant because (as the authors point out), studies in primary care are far to scarce. The aim of the authors is very important and results collect are relevant. That is why I suggested to accept this article. But this article does not provide enough background for an international audience, gives too much details on some aspects and unnecessarily repeats some less relevant information. It is unclear why this study is prospective as I had the impression a clinical program has been studied afterwards (see below: several paragraphs are unclear about this). This article needs a major revision before it is suitable to BMJ Open. Some suggestions: Title: why did the authors add "A primary care approach to the COVID-19 pandemic:" in front of the title? To me it seems as though this article will present an approach to a patient with COVID-19 or a public health approach but this article merely describes this disease in primary care. This title is not conform STROBE-guidelines. Abstract: -methods: the aim of this program is unclear: did the authors study a program designed for medical care or did they design a platform for research only? conclusion: why not change "some symptoms" to those mostly associated with a positive diagnosis? -Strengths and limitations: 1. "Necessarily" is not correct although I understand what the authors mean Introduction -Brazil is hit rather hard by COVID-19, using the available evidence the authors should reflect on this national or local epidemiology: why is it hit so hard? What are the strengths and weaknesses of the local healthcare system? What was the testing strategy and was it carried out well? Please add the very basics of primary care organization in Brazil including payment model and population coverage.
---

	- P7L13: others followed by only one reference is not correct, more references should be added in order to make this statement -P7L20: same problem, the WHO is not the only relevant source of information. Are the authors aware of any systematic reviews? -P7L46 there are some primary care studies available so it would be interesting to adjust this paragraph with the little available information. Please notice some articles have been published very recently. Settings -The active ageing index does not seem relevant and makes this paragraph hard to read. For me it is sufficient to know the population is ageing and rather rich/educated compared to other Brazilian regions. -Corona São Caetano Platform: The aim of this platform is not clear: merely research or also clinical care? In the latter case I would suggest to give some information about the medical care provided. Does this platform collaborate with the patient's own GP or is this GP biased in case of suspected COVID-19? Does it prescribe drugs? Social support? Treatment of co-morbidities, ...? P8L28: how were patient "encouraged"? P8L36: please add a reference: who made this case definition? P8L58: please avoid the passive voice wherever possible: I understand the patient collected it's own sample? Why did was this procedure chosen? The professionals did wear protection anyhow. P9L10: this sentence is not correct. Follow-up procedures: again, the aim of these procedures is unclear. Why medical students? P9L31: please add a reference so an interested reader can find out the rationale for these 14 days. Study dates: this paragraph should be more concise and readable and not contain the same information twice. Statistical methods: This section should also be reviewed by a specialist in methodology/virology! I do not know whether the proposed methods for handling the large amount of missing values is correct. P10L17: I do not understand this sentence. P10L27: it is not necessary to repeat the follow up period. P10L14: how did the authors link data? Using a national number? Ethics: see previous remark: how did the authors link several data sources without compromising anonymity? Patient involvement: this is surely a weakness of this study Results I would suggest to start with a description of the studied population (table 1) in stead of the results themselves. Many of the presented results are statistically significant but clinically irrelevant. Epidemiological and programmatic indicators P12L12: the study period is repeated again P12L34: who diagnosed the other 12% Does the database include all cases? Why did the authors only adjust for age, sex; delay froms symptom onset and PCR platform used. Table 1 and 2 reveal many more possible confounders/co-variates. Discussion -P15L31: 18% does not seem lower then 20% from a clinical perspective. Confidence-intervals? -P15L38: the mean delay for swabbing was 5 days, why is it so long? What is the delay between first telephone contact and swabbing? Is this a patient's delay or a system delay? Does this delay influence the rate of false negatives? As this rate increases
--	---

	with a longer delay.  -Please compare the results more often with systematic reviews. How do the symptoms compare to other studies (mostly hospital care) -the introduction to Brazilian healthcare should be transferred to the introduction and updated as suggested above -I do not understand the suggestion to use the same approach for other diseases as this article presents an interesting insight into COVID-19 but does not prove the usefulness of the program as no outcomes were measured neither compared to other approaches. -There is no final conclusion, ending the paper with this dubious statement seems rather strange. -P16L45: 100% coverage means all inhabitants have a family doctor? Seems unrealistic? Homeless people? People without a legal status? The discussion does not include all key results (Strobe item 18) and does not include all weaknesses (item 19). Some points I have missed in the discussion:  -Chronic respiratory disease was less frequent in RT-PCR positive than dual-negative patients.=> why? -More cough in RT pos patients: more virus in the nose? -Spreading of the CT-values: there is a regressionline but given the spread I do not see any clinical relevance. -28% is a high positivity rate, due to local testing policy? Low rate of influenza or other infections during the study time? High spread of the disease in the community? Please compare it to the positivity rate of other outbreaks (US, Italy, Japan, China, ...) -Those with a higher education were significantly less often positive: social class effect? -Anosmia of 30% in those testing double negative seems very high as in my experience this symptom is seen rarely in other diseases. Anosmia means no smell at all or did the authors also include a reduced ability to smell? - Reason for hospitalisation RTPCR-negative patients: because of COVID-19 or because of another reason unrelated to COVID-19? CONTRIBUTION STATEMENT: It seems unlikely that all these authors actually meet the criteria for authorship. Providing clinical oversight and supervision of students alone is not enough to be a co-author neither is the collection and curation of data. CONFLICTS OF INTEREST STATEMENT there are conflicts of interests: working in a program and studying it at the same time is a conflict of interest. Didn't any of the authors get paid for this clinical work? Each author should fill in a ICMJE Conflict of Interest form FUNDING STATEMENT this is not clear enough, again no distinction between a clinical program and a research project DATA SHARING STATEMENT this is a weak data sharing statement: some of the data should be made available. I understand the results of the interviews are confidential but the data from the PCR-testing machines? Please motivate why data is not available.
--	--

REVIEWER	Dominique Costagliola Institut Pierre Louis d'Epidémiologie et de Santé Publique, Sorbonne Université, INSERM
REVIEW RETURNED	04-Oct-2020

GENERAL COMMENTS	General comment There is no clear research objective for this paper, apart from describing the Corona Sao Caetano program. Given that, it is
---

	unclear which part of this descriptive report may be extrapolated or contribute to other programs. Many directions have been explored, but none conducted to a clear message for the readers. I made some suggestions below. Detailed comments It is unclear why the study period was restricted to one month, given that a larger study would be more powerful. There are large number of participants not evaluated at each step. The impact of this fact on the described associations with symptoms or with the number of CT should be accounted for when discussing the results. It is unclear why analyses are mainly univariable analyses, and why the only multivariable analysis is conducted to assess parameters linked with the number of CT at diagnosis? It would have been interesting to assess the most pertinent combinations of symptoms or cluster of symptoms with the COVID-19 status, rather than to perform only univariable analysis. I do not understand why it is interesting to assess the parameters associated with the numbers of CT at diagnosis ? has this any practical interest? Is not the strongest factor the delay between onset of symptoms and nasal sampling time, with almost no change of the association after adjustment? It is also unclear whether the relationship with age was linear On the other hand, multivariable analyses of factors associated with being tested PCR positive or of factors associated with severe diseases, both on a larger sample would have been interesting (see for instance Reilev M et al, Int J Epidemiol 2020). Legend of Figure S3 right panel patients not patientes Why are they 2 different graphs on figure S4 as the right panel is informative enough ? What is the interest of Figure S5, individual graphs, while the right panel may be used to assess a duration of symptoms Interpreting PCR-antibodies+, as false negative is not fully justified and the percentage is an underestimation, given the large untested proportion.
--	--

VERSION 1 – AUTHOR RESPONSE

Reviewer: 1

General remarks:

This article with a simple design is relevant because (as the authors point out), studies in primary care are far too scarce. The aim of the authors is very important and results collected are relevant. That is why I suggested to accept this article.

Reply. thank you for your positive comments

But this article does not provide enough background for an international audience, gives too much details on some aspects and unnecessarily repeats some less relevant information.

Reply. We hope to have been able to address the concerns raised. See our point-by-point response below.

Query 1.1 It is unclear why this study is prospective as I had the impression a clinical program has been studied afterwards (see below: several paragraphs are unclear about this).

Reply. The collection of patient information – in particular demographics and symptoms – was prospective in relation to the PCR test. This information was collected in a standardized, pre-specified way. We use the term “prospective” to distinguish from an alternative study design where these data are acquired retrospectively, such as a chart review.

Query 1.2 This article needs a major revision before it is suitable to BMJ Open. Some suggestions:

Title: why did the authors add "A primary care approach to the COVID-19 pandemic:" in front of the title? To me it seems as though this article will present an approach to a patient with COVID-19 or a public health approach but this article merely describes this disease in primary care. This title is not conform STROBE-guidelines.

Reply. We have amended the title as follow “Clinical features and natural history of the first 2,073 suspected COVID-19 cases in the Corona São Caetano primary care programme: a prospective cohort study”

Query 1.3 Abstract:

-methods: the aim of this program is unclear: did the authors study a program designed for medical care or did they design a platform for research only?

Reply. The Corona São Caetano programme was designed with both objectives in mind, but its primary goal was the provision of community-based clinical care for all residents of São Caetano do Sul with symptoms suggestive of COVID-19. As we note in the manuscript, it is challenging to establish community cohorts in a rapidly evolving pandemic. As such, we specifically planned the programme so that patient data was collected in a standardized way, with pre-defined follow-up, testing, and sample storage for later evaluation. See clarification in the section “Corona São Caetano Platform:

“The objective of the platform was to offer clinical care for patients with flu syndrome and suspected COVID-19. Through the multimedia platform, patients could be triaged and guided in relation to their clinical needs and tested, without having to leave their homes or go to health facilities, unless seriously ill. This strategy aimed at reducing the workload in health units and the risk of SARS-CoV-2 transmission in the population served by these health units. Patients’ GPs were informed of lab results and had access to clinical data stored in the platform. GPs were expected to call patients being assisted by the platform and provide medical assistance through home visits or at the primary care clinic if needed. In general, the drugs prescribed through the platform were restricted to analgesics and antipyretics. The platform was designed so that clinical information was collected in a standardized way for research purposes.”

Query 1.4 conclusion: why not change "some symptoms" to those mostly associated with a positive diagnosis?

Reply. We have amended as requested.

Query 1.5 Strengths and limitations: 1. “Necessarily” is not correct although I understand what the authors mean

Reply. “Necessarily” is indeed redundant in this sentence, and we have removed it.

Query 1.6 Introduction

-Brazil is hit rather hard by COVID-19, using the available evidence the authors should reflect on this national or local epidemiology: why is it hit so hard?

Reply. The question of why Brazil was so hard-hit by the SARS-CoV-2 pandemic is complex, not least given the continental scale and vast regional differences in governance and socioeconomic conditions. We do not believe this topic is germane to our paper. A discussion of this issue is not required for the reader to follow the content of the manuscript.

Query 1.7 What are the strengths and weaknesses of the local healthcare system? What was the testing strategy and was it carried out well? Please add the very basics of primary care organization in Brazil including payment model and population coverage.

Reply. Please see the following addition to the introduction:

“Primary health care (PHC) in Brazil is provided by the publicly funded Unified Health System (SUS – Portuguese acronym) within the family health strategy (*Estratégia Saúde da Família*). Provision of care is centred around a healthcare unit with a multi-professional team that is responsible for all residents in the immediate catchment area⁸. Nearly two-thirds of the Brazilian population is covered by the family health strategy⁸.”

Query 1.8 - P7L13: others followed by only one reference is not correct, more references should be added in order to make this statement

Reply. We have added additional references as requested.

Query 1.9 -P7L20: same problem, the WHO is not the only relevant source of information. Are the authors aware of any systematic reviews?

Reply. We have added additional references as requested.

Query 1.10 -P7L46 there are some primary care studies available so it would be interesting to adjust this paragraph with the little available information. Please notice some articles have been published very recently.

Reply. Please see amended paragraph and additional references.

Query 1.11 Settings

-The active ageing index does not seem relevant and makes this paragraph hard to read. For me it is sufficient to know the population is ageing and rather rich/educated compared to other Brazilian regions.

Reply. We have removed the detail on the active ageing index to improve readability, as suggested.

Query 1.12 -Corona São Caetano Platform: The aim of this platform is not clear: merely research or also clinical care? In the latter case I would suggest to give some information about the medical care provided. Does this platform collaborate with the patient's own GP or is this GP biased in case of suspected COVID-19? Does it prescribe drugs? Social support? Treatment of co-morbidities, ...?

Reply. Please see response to query 1.3

Query 1.13 P8L28: how were patient "encouraged"?

Reply. The programme was publicized initially through local media reports. See clarification to line:

“Residents of the municipality aged 12 years and older with suspected COVID-19 symptoms were encouraged, through local media reports, to contact the dedicated Corona São Caetano platform via the website (access at <https://coronasaocaetano.org/>) or by phone.”

Query 1.14 P8L36: please add a reference: who made this case definition?

Reply. This case definition was developed by the authors FEL, MCMC and RB - infectious disease specialist clinicians with primary care experience. It was developed to encompass the known symptoms of COVID-19 at the time the programme was being developed (March-April 2020) and is similar to the national Brazilian case definition. See clarification”

“The case definition was developed in consultation with infectious disease and primary care specialists to encompass the known symptoms of COVID-19 and is similar to the Brazilian national case definition²⁰.”

Query 1.15 P8L58: please avoid the passive voice wherever possible: I understand the patient collected it's own sample? Why did was this procedure chosen? The professionals did wear protection anyhow.

Reply. We have amended to replace with the active voice.

The reviewer is correct that we used a self-collection procedure as mentioned in the text under the section "Sample collection": "Patients self-collected nasopharyngeal swabs (NPS – both nostrils and throat) at their own homes under the supervision of trained healthcare personnel."

Self-collection of nasopharyngeal swabs for the molecular detection of SARS-CoV-2 has been recommended as an alternative validated method of collection for samples from patients with suspected COVID-19, as well as other respiratory diseases (e.g., Akmatov et al. PlosOne 2012 e48508), and has the considerable advantage of reducing the chance of aerosol transmission to healthcare professionals. As already explained in the manuscript, one of the objectives of the São Caetano platform was to reduce the transmission of SARS-CoV-2 in the population of the municipality of São Caetano do Sul, which includes health professionals working at the health units in the city.

Thus, we opted for the use of this type of collection, in order to avoid the need for patients to go to health units and to reduce the risk of exposure of health professionals who went to patients' homes. Please see further clarification under the "Sample collection" manuscript section.

The health professionals did wear PPE, as is appropriate given they were having clinical contact with patients with suspected COVID-19.

Query 1.16 P9L10: this sentence is not correct.

Reply. We believe this sentence is correct. This type of comment – where the specifics of what the reviewer requires is not stated – is difficult to respond to. Could the reviewer provide more information about what error they have identified?

Query 1.17 Follow-up procedures:
again, the aim of these procedures is unclear. Why medical students?

Reply. We have added the following clarification "The purpose of the follow-up was to assess clinical evolution. Where patients were judged to be deteriorating or developing severe disease they were signposted to secondary care services."

Final year medical students – under appropriate supervision – were recruited to administer the remote consultations. Due to the disruption caused by the evolving COVID-19 epidemic, the medical school's normal clinical activities were partly suspended. Participating in the Corona Platform was therefore both an opportunity for learning and service provision. This is addressed in the Discussion section.

Query 1.18 P9L31: please add a reference so an interested reader can find out the rationale for these 14 days.

Reply. The average time of excretion of this virus for patients with mild to moderate disease is roughly 14 days, which can be extended for 20-30 days and exceptionally up to 90 days. We have added a reference as requested.

Query 1.19 Study dates: this paragraph should be more concise and readable and not contain the same information twice.

Reply. We have shortened this paragraph to improve its clarity.

Query 1.20 Statistical methods:

This section should also be reviewed by a specialist in methodology/virology! I do not know whether the proposed methods for handling the large amount of missing values is correct.

Reply. We used a standard analytic approach. We used commonly used tests of statistical significance.

There was not a large amount of missing values - see table legends that specify the number of missing values. This is a strength of our platform. Because the data collection process was standardized, the level of data completion was high. It is therefore unlikely that this is an important threat to the validity of our findings. If the reviewer can highlight a particular issue that they have identified, then we would be happy to respond to that.

Query 1.21 P10L17: I do not understand this sentence.

Reply. Please see the amendment at this line in the revised document.

Query 1.22 P10L27: it is not necessary to repeat the follow up period.

Reply. We have removed this minor repetition.

Query 1.23 P10L14: how did the authors link data? Using a national number?

Ethics: see previous remark: how did the authors link several data sources without compromising anonymity?

Reply. The data were linked separately by the author SRPS who did not have access to the full analytic dataset. This author searched the SIVEP-Gripe system and the municipal epidemiological surveillance dataset using full name and date of birth and assigned the hospitalization or mortality status to the corresponding study ID. Please see the following clarification to the manuscript”

“Linkage was last performed on 5th June 2020, 23 days after the last patient was enrolled, by the author SRPS who did not have access to the full analytic dataset. This author searched the SIVEP-Gripe system and the municipal epidemiological surveillance dataset using full name and date of birth”

Query 1.24 Patient involvement: this is surely a weakness of this study

Reply. We agree with this observation and note this weakness in the discussion. However, it is important also to note that the Corona Platform was developed in the beginning of the SARS-CoV-2 pandemic, under unprecedented time and resource pressure. The involvement of patients in research planning is clearly challenging under these circumstances.

Query 1.25 Results

I would suggest to start with a description of the studied population (table 1) in stead of the results themselves. Many of the presented results are statistically significant but clinically irrelevant.

Reply. The reason for beginning results section in this way is that we wish to provide an overview of the program indicators and patient flow, before going onto discuss the study population. This is a standard way of presenting results. This ordering is a style decision and we do not believe that it impacts on the reader’s understanding nor the rigor of the study.

We agree with the importance of emphasizing clinical over statistical significance. However, it is not clear which results the reviewer is referring to, as they have only stated that “many” results are in this category of purely statistical significance. If they can list the cases in which we have over-interpreted results where the effect size is small then we will be happy to amend as appropriate.

Query 1.26 Epidemiological and programmatic indicators
P12L12: the study period is repeated again

Reply. We have altered this repetition from “Between 13th April and 13th May 2020” to “Over the study period”.

Query 1.27 P12L34: who diagnosed the other 12% Does the database include all cases?

Reply. Presumably the reviewer is referring to this phrase

“At the beginning of programme role out, 75% of notified COVID-19 cases in São Caetano do Sul were diagnosed in outpatient or hospital services. Over the study period, adherence to the programme increased, and by May 13th, 2020, 78% of cases in the municipality were diagnosed within our programme.”

It is unclear which 12% the reviewer is referring to as $100\% - 78\% = 22\%$, and $100\% - 75\% = 25\%$. The way to interpret this is that 25% of cases were diagnosed within the programme at the beginning of the period, increasing to 78% by the end. Please see clarification to the text:

“At the beginning of programme role out, 25% of notified COVID-19 cases in São Caetano do Sul were diagnosed in our programme. Over the study period, adherence to the programme increased, and by May 13th, 2020, this figure had risen to 78%.”

Query 1.28 Why did the authors only adjust for age, sex; delay forms symptom onset and PCR platform used. Table 1 and 2 reveal many more possible confounders/co-variates.

Reply. The variables age, sex and (most importantly) time from symptom onset are known to be associated with Ct number. The other variables in the tables (educational level, BMI, occupation etc.) are not. Furthermore, as we explain in the text, the model was built to assess the association between symptoms and Ct number. Age, sex and time from symptom onset are associated with both the reported symptoms (see results) as well as Ct, and therefore are expected to act as confounders.

Query 1.29 Discussion

-P15L31: 18% does not seem lower than 20% from a clinical perspective. Confidence-intervals?

Reply. We agree. A more appreciate wording is

“This is similar to a pooled analysis showing a false-negative rate for RT-PCR of 20% at three days post-symptom onset.²⁸”

We have changed the text accordingly.

Query 1.30 -P15L38: the mean delay for swabbing was 5 days, why is it so long? What is the delay between first telephone contact and swabbing? Is this a patient's delay or a system delay? Does this delay influence the rate of false negatives? As this rate increases with a longer delay.

Reply. This delay is predominantly the time for patient to contact the platform, i.e. time from symptom onset to first phone call/website visit. The median delay from first phone call to swab was one day (IQR 1-2 days). We have added this information to the Section “**Associations between SARS-CoV-2 RT-PCR Cycle threshold (Ct) values, and demographic and clinical features**”. As noted, the delay between symptom onset and swab collection does influence the rate of PCR false negatives (Kucirka, Ann Int Med doi:10.7326/M20-1495).

Query 1.31 -Please compare the results more often with systematic reviews. How do the symptoms compare to other studies (mostly hospital care)

Reply. Please see the addition of reference 31: a recent systematic review of diagnostic features of COVID-19 in the emergency department.

Query 1.32 -the introduction to Brazilian healthcare should be transferred to the introduction and updated as suggested above

Reply. Please see the amendment to the introduction as suggested and response to query 1.7.

Query 1.33 -I do not understand the suggestion to use the same approach for other diseases as this

article presents an interesting insight into COVID-19 but does not prove the usefulness of the program as no outcomes were measured neither compared to other approaches.

Reply. We agree that the reviewer is correct to highlight this point and this conclusion was overstated. We have made some qualifying statements as follows:

“But we believe that this infrastructure could be implemented in other regions with less resources. Other respiratory disease such as influenza, measles, or tuberculosis may benefit from similar approach. However, further evaluation of the impact of the Corona Platform are required.”

Query 1.34 -There is no final conclusion, ending the paper with this dubious statement seems rather strange.

Reply. Please see the addition of a final conclusion.

Query 1.35 -P16L45: 100% coverage means all inhabitants have a family doctor? Seems unrealistic? Homeless people? People without a legal status?

Reply. There are enough GP practice to cover 100% of the area of the city. Homeless people have access to healthcare through different strategies more appropriate to socially vulnerable populations. São Caetano provided free healthcare in field hospitals for COVID-19 cases among homeless people and citizens who were unable to perform appropriate self-care or self-isolation, regardless of clinical severity.

Query 1.36 The discussion does not include all key results (Strobe item 18) and does not include all weaknesses (item 19).

Reply. Please see our response to the points raised below.

Query 1.37 Some points I have missed in the discussion:

-Those with a higher education were significantly less often positive: social class effect?

Reply. Please see following addition to discussion:

“The profile of suspected cases that tested positive for COVID-19 differed in some important respects from those testing negative. The lower educational level among positive cases suggests that, in São Caetano do Sul, the risk of exposure to COVID-19 follows a socioeconomic gradient, consistent with other findings from Brazil^{25,26}. Although more women presented to the platform, proportionally more men tested positive, consistent with data from São Paulo showing a higher seroprevalence in men than women²⁷, but also potentially reflecting different health seeking behaviours. Comorbidities were mostly similar, although chronic respiratory disease was less frequent in those testing RT-PCR positive. This may be due to a proportion of presentations in those with chronic respiratory disease being explained by exacerbations of their underlying pathology from aetiologies other than SARS-CoV-2, as well as higher anxiety about COVID-19 in those with pre-existing respiratory disease.”

Query 1.38 -Chronic respiratory disease was less frequent in RT-PCR positive than dual-negative patients.=> why?

Reply. Please see response to query 1.37.

Query 1.39 -More cough in RT pos patients: more virus in the nose?

Reply. Presumably the reviewer is referring to the difference in the frequency of reported cough in RT-PCR positive versus seropositive patients: 77% and 63%, respectively. We do not think this is a key point to highlight in the discussion. The 95% confidence intervals overlap (see Figure 2A) and the difference is relatively small.

Query 1.40 -Spreading of the CT-values: there is a regression line but given the spread I do not see any clinical relevance.

Reply. We agree with the reviewer's comment and the following text to the discussion:

"As expected, the main determinant of Ct was the delay between symptom onset and swab collection, mostly due to the delay in reporting to the platform. After adjusting for this, as well as age and sex, we found that a self-reported fever and arthralgia were associated with lower Cts. The presence of these symptoms may identify patients with a higher viral load in the community. However, these results should be seen as purely exploratory, and the wide spread of Ct values around the regression line precludes a direct clinical application at present."

Query 1.41 -28% is a high positivity rate, due to local testing policy? Low rate of influenza or other infections during the study time? High spread of the disease in the community? Please compare it to the positivity rate of other outbreaks (US, Italy, Japan, China, ...)

Reply. It is not possible to compare directly with other countries. The PCR positivity rate in such a primary care testing program will depend on many factors, including the magnitude of the epidemic in a given region; at what time in relation to the epidemic peak the programme is being performed; the case definition to be eligible for testing; the co-circulation of other respiratory viruses, not only influenza viruses but also other season coronaviruses, rhinovirus etc. For this reason, we have chosen not to add this to the discussion.

Query 1.42 -Anosmia of 30% in those testing double negative seems very high as in my experience this symptom is seen rarely in other diseases. Anosmia means no smell at all or did the authors also include a reduced ability to smell?

Reply. We agree. It is our experience as well that 30% anosmia does appear to be high in the double negative patients. However, this is not a symptom that is perhaps routinely asked about – at least not prior to the COVID-19 epidemic – as it has not previously carried a great deal of diagnostic importance. Our results show that when we systematically asked about anosmia the prevalence was high. The reviewer is correct that there was not a strict definition – i.e. using a validated assessment tool – of anosmia, and this number will include some cases of hyposmia, as opposed to true complete loss of smell.

We would expect a very small number of false-negative PCR and false-negative serology, but nowhere near 30%.

Please see the following addition to the text:

"This is consistent with systematic review evidence highlighting anosmia and ageusia as key diagnostic features of COVID-19³¹. It is of note that 30% of jointly RT-PCR and serology negative patients reported these symptoms, indicating that although indicative of COVID-19, the specificity of these symptoms is not high enough to rule in the diagnosis alone"

Query 1.43 - Reason for hospitalisation RTPCR-negative patients: because of COVID-19 or because of another reason unrelated to COVID-19?

Reply. RT-PCR negative patients were hospitalized due to an acute respiratory syndrome of an etiology other than SARS-CoV-2. Please see the clarification:

"The rate of hospitalization was lower (0.5%) in those testing PCR-negative. These patients were admitted with a severe acute respiratory syndrome of an aetiology other than SARS-CoV-2. The 14-fold higher admission rate among PCR-positive cases highlights the importance of molecular testing for SARS-CoV-2 in patients presenting with features of respiratory viral illness to primary care."

Query 1.44 CONTRIBUTION STATEMENT: It seems unlikely that all these authors actually meet the criteria for authorship. Providing clinical oversight and supervision of students alone is not enough to be a co-author neither is the collection and curation of data.

Reply. "It seems unlikely that all these authors actually meet the criteria for authorship." It is not clear to us how the reviewer has arrived at this conclusion. All authors contributed to the intellectual content

of the manuscript in weekly lab meetings and reviewing and revising the final manuscript. This is detailed in the contribution statement as well as the other specific roles that the co-authors fulfilled.

Query 1.45 CONFLICTS OF INTEREST STATEMENT there are conflicts of interests: working in a program and studying it at the same time is a conflict of interest. Didn't any of the authors get paid for this clinical work? Each author should fill in a ICMJE Conflict of Interest form

Reply. The platform took advantage of some existing systems: public health system (municipal health department of São Caetano do Sul , which funded the establishment and implementation of the platform), medical students from a private university and partnership with a public university. The authors did not get paid to participate: all were regular employees of the different participating institutions.

As such, we do not understand this to be a conflict of interest, although we are willing to include within the conflict of interest statement that some authors were involved in providing clinical care within the programme.

Query 1.46 FUNDING STATEMENT this is not clear enough, again no distinction between a clinical program and a research project

Reply. We have addressed this issue in query 1.3.

Query 1.47 DATA SHARING STATEMENT this is a week data sharing statement: some of the data should be made available. I understand the results of the interviews are confidential but the data from the PCR-testing machines? Please motivate why data is not available.

Reply. We have amended our data sharing statement. De-identified data underlying the main analyses will be released on a linked Figshare repository upon acceptance of the manuscript.

Reviewer: 2

Query 2.1 General comment

There is no clear research objective for this paper, apart from describing the Corona Sao Caetano program. Given that, it is unclear which part of this descriptive report may be extrapolated or contribute to other programs. Many directions have been explored, but none conducted to a clear message for the readers. I made some suggestions below.

Reply. We respectfully disagree with the reviewer that the objectives were not clear. We describe them in the introduction as:

“to describe the epidemiological indicators of the early phase of the programme rollout [1] ; and to describe the clinical [2], virologic [3] and natural history features (including hospitalization and deaths) [4] of SARS-CoV-2 infection among patients identified in primary care.”

We agree the objectives are primarily descriptive. However, we believe that we addressed all the aims in the manuscript:

- [1] The epidemiological indicators include the number of daily presentations, the positivity rate, the engagement with follow-up etc.
- [2] Clinical presenting features are described and compared with PCR and antibody negative controls. Symptom evolution in PCR+ve individuals is reported. Symptom frequency according to sex is also shown
- [3] We provide an analysis of the clinical and demographic correlates of the nasopharyngeal viral load at presentation
- [4] Hospitalization and case fatality rates are reported, addressing to the natural history

Query 2.2 Detailed comments

It is unclear why the study period was restricted to one month, given that a larger study would be more powerful.

Reply. We agree with this observation: a larger study would be more powerful. The decision to finalize the analytic dataset after one month, analyze and release the results as a pre-print (DOI <https://doi.org/10.1101/2020.06.23.20138081>) was influenced by the urgency to provide information on community COVID-19 cases in the evolving pandemic. At the time we were not aware of any work on this issue. The nature of this type of study – marrying community care with research – is that the dataset could always be expanded by waiting longer. We had to make a somewhat arbitrary decision about when to finalize the study period. This does not preclude the possibility of using the data from the São Caetano platform to answer questions that require a larger sample size, some of which the reviewer has correctly suggested.

We have amended the title to reflect the fact our paper presents results from the early part of the São Caetano platform: “Clinical features and natural history of the first 2,073 suspected COVID-19 cases in the Corona São Caetano primary care programme: a prospective cohort study”

Query 2.3 There are large number of participants not evaluated at each step. The impact of this fact on the described associations with symptoms or with the number of CT should be accounted for when discussing the results.

Reply. Thank you for highlighting this important point. Please see the following discussion of this issue in the manuscript:

“Firstly, serology was not performed on all RT-PCR negative patients due to on-going symptoms, loss to follow-up, or patient refusal. Of note, none of the RT-PCR-negative patients that were admitted to hospital underwent serology testing. This suggests that patients who were not tested with serology may have had a higher prevalence of COVID-19 than those that were tested. In addition, imperfect serology test performance (81% sensitivity)²³ will introduced false-negative results. Taken together, these biases may have underestimated the true seroprevalence among RT-PCR-negative cases, as well as the false-negative rate of RT-PCR. The latter calculation may also have been influenced by the inclusion of RT-PCR positive patients in the denominator, introducing an incorporation bias.³⁶ Furthermore, the association between symptoms and COVID-19 diagnosis was based on the comparison with doubly PCR and serology negative individuals. It is not clear how the exclusion of individuals that did not undergo serology testing would have influenced these associations.”

Query 2.4 It is unclear why analyses are mainly univariable analyses, and why the only multivariable analysis is conducted to assess parameters linked with the number of CT at diagnosis?

Reply. Very few patients were hospitalized, and even fewer died. The small number of individuals with these outcomes meant that a multivariate analysis to determine risk factors for these outcomes was not possible. We have clarified this issue in the methods section as follows “A multivariate analysis was not conducted due to the small number of individuals experiencing this outcome [death or hospitalization] .”

Query 2.5 It would have been interesting to assess the most pertinent combinations of symptoms or cluster of symptoms with the COVID-19 status, rather than to perform only univariable analysis.

Reply. We agree this would be an interesting analytic approach. However, it is our understand that unsupervised machine learning / cluster identification relies on a large sample size, and the present dataset is not large enough to support this approach. This is something we are looking into doing as the programme accrues further data.

Query 2.6 I do not understand why it is interesting to assess the parameters associated with the numbers of CT at diagnosis ? has this any practical interest? Is not the strongest factor the delay between onset of symptoms and nasal sampling time, with almost no change of the association after

adjustment? It is also unclear whether the relationship with age was linear. On the other hand, multivariable analyses of factors associated with being tested PCR positive or of factors associated with severe diseases, both on a larger sample would have been interesting (see for instance Reilev M et al, Int J Epidemiol 2020).

Reply. Different authors have found a positive association between SARS CoV2 viral load and disease severity as well as risk of progression among hospitalized patients. Therefore, we felt it was of interest to explore any possible association between different clinical manifestations (specifically symptoms at presentation) and viral load among patients with mild to moderate COVID-19. We are not aware of published data on this topic. It is for this reason that we included the analysis of cycle thresholds according to the presence of each symptom at presentation.

The most important factor influencing cycle threshold is the time from symptom onset to swab collection, hence its inclusion in our model. We agree with the reviewer that the apparent associations between reported symptoms and viral load are partially explained by this variable. Please see the following addition to the discussion section:

“As expected, the main determinant of Ct was the delay between symptom onset and swab collection, mostly due to the delay in reporting to the platform. After adjusting for this, as well as age and sex, we found that a self-reported fever and arthralgia were associated with lower Cts. The presence of these symptoms may identify patients with a higher viral load in the community. However, these results should be seen as purely exploratory, and the wide spread of Ct values around the regression line precludes a direct clinical application at present.”

The relationship between age and cycle threshold was linear. Furthermore, we performed the appropriate tests of model assumptions.

We agree with the observation that a multivariate analysis to establish factors associated with hospitalization and death would be of interest. Unfortunately, the small number of individuals experiencing these outcomes in our cohort meant we did not have sufficient power to conduct these analyses. This would be a different paper.

Query 2.7 Legend of Figure S3 right panel patients not patients

Reply. Thank you for highlighting this typo. We have amended.

Query 2.8 Why are they 2 different graphs on figure S4 as the right panel is informative enough ?

Reply. We agree with this observation. Please see amended Figure S4.

Query 2.9 What is the interest of Figure S5, individual graphs, while the right panel may be used to assess a duration of symptoms

Reply. We agree there is some redundancy here. The interest of the individual graphs is to show the underlying data structure. The right-hand panel is a summary. As this is a supplemental figure we believe the extra information is valuable to the reader.

Query 2.10 Interpreting PCR-antibodies+, as false negative is not fully justified and the percentage is an underestimation, given the large untested proportion.

Reply. We agree this is an important limitation. We have assumed the proportion testing seropositive among the PCR-ve patients that did not undergo serology testing was the same as among those that did (8.6%). If those that did not get an antibody test were systematically different from those that did – i.e. more or less likely to have been truly a COVID-19 case – then this would bias our estimate of the PCR false-negative proportion. We acknowledge this as a limitation as follows:

“Of note, none of the RT-PCR-negative patients that were admitted to hospital underwent serology testing. This suggests that patients who were not tested with serology may have had a higher prevalence of COVID-19 than those that were tested. In addition, imperfect serology test performance

(81% sensitivity)²⁰ will introduced false-negative results. Taken together, these biases may have underestimated the true seroprevalence among RT-PCR-negative cases, as well as the false-negative proportion for RT-PCR.”

VERSION 2 – REVIEW

REVIEWER	Stefan Morreel University of Antwerp, Belgium
REVIEW RETURNED	28-Nov-2020

GENERAL COMMENTS	The authors have made impressive efforts to improve the paper. I have some remaining very minor remarks: Query 1.6: I still believe that the question of why Brazil was hit so hard should be mentioned. This can be done very briefly by adding on sentence and a reference as there are probably some papers available about this topic. I agree this issue should not be discussed but readers unaware of the Brazilian situation have to be informed about it Query 1.12: it has been made much clearer how the program works. The link leads to a page. Suggestion: add some printscreens to the supplementary material. Other primary care initiatives worldwide might be interested how this multimedia platform works but cannot access it as the provided URL requires a code. Query 1.20 it is up to the editor to judge whether extensive statistical review is necessary, I can only state that this is not my field of expertise. Query 1.25: I meant the list of symptoms of COVID-19: the differences between RT-PCR and seropositive patients were rather small. It is fine by me leave this as it is now. Query 1.27: because of the improvements made above, this remark is no longer relevant Because I now understand what this program was about I think the authors should briefly address two more weaknesses: recruitment by local media (selection bias?), use of a multi-media platform (patients without the necessary ICT knowledge/resources might not be able to use the platform).
--

VERSION 2 – AUTHOR RESPONSE

Query 1.6: I still believe that the question of why Brazil was hit so hard should be mentioned. This can be done very briefly by adding on sentence and a reference as there are probably some papers available about this topic. I agree this issue should not be discussed but readers unaware of the Brazilian situation have to be informed about it

Reply. We agree this would be useful to the interested reader. We have added some references on this issue and updated the text:

“In Brazil, the first case of COVID-19 was identified in the city of São Paulo on 26th February 2020.⁹ As of 1st Dec 2020 there were over 6 million confirmed cases nationally, with São Paulo contributing a fifth of these.¹⁰ The reasons for the exceptionally large epidemic in Brazil have been discussed elsewhere^{11–13}.”

Query 1.12: it has been made much clearer how the program works. The link leads to a page. Suggestion: add some printscreens to the supplementary material. Other primary care initiatives worldwide might be interested how this multimedia platform works but cannot access it as the provided URL requires a code.

Reply. Thank you for this good suggestion. We have added some screenshots to the SM and removed the link as it requires a local post code to access.

Query 1.25: I meant the list of symptoms of COVID-19: the differences between RT-PCR and seropositive patients were rather small. It is fine by me leave this as it is now.

Reply. Thank you for the clarification.

Query 1.27: because of the improvements made above, this remark is no longer relevant. Because I now understand what this program was about I think the authors should briefly address two more weaknesses: recruitment by local media (selection bias?), use of a multi-media platform (patients without the necessary ICT knowledge/resources might not be able to use the platform).

Reply. Thank you for these helpful additional comments.

The residents of São Caetano were initially made aware of the existence of the Corona platform, in part, by local media reports. With respect to selection bias, it is conceivable that people that consume local media may have been overrepresented in the early phase. Although an interesting observation, it is not clear to us how such a selection bias would impact the external validity of our study.

We anticipated the issue of excluding residents without sufficient ICT knowledge/resources, and for that reason a phone line/call center was available as an alternative point of contact. During the study period, half the initial contacts with the system were via telephone call. We have amended the sentence in the methods, as follows:

“Through the multimedia platform (website of phone call), patients could be triaged and guided in relation to their clinical needs and tested, without having to leave their homes or go to health facilities, unless seriously ill”

We have also amended the first sentence of the results:

“Over the study period, there were 2,073 presentations (49% phone call, 51% website), from 2,011 individual patients, that met the criteria for a suspected COVID-19 case (See Figure 1 for study flow).”

VERSION 3 – REVIEW

REVIEWER	Stefan Morreel University of Antwerp
REVIEW RETURNED	04-Dec-2020
GENERAL COMMENTS	Thank you for these last changes. Minor remark:“(website of phone call)” should be website OR phone call?